# Online Posterior Sampling with a Diffusion Prior

**Branislav Kveton**
Adobe Research*

**Boris N. Oreshkin**
Amazon

**Youngsuk Park**
AWS AI Labs

**Aniket Deshmukh**
AWS AI Labs

**Rui Song**
Amazon

## Abstract

Posterior sampling in contextual bandits with a Gaussian prior can be implemented exactly or approximately using the Laplace approximation. The Gaussian prior is computationally efficient but it cannot describe complex distributions. In this work, we propose approximate posterior sampling algorithms for contextual bandits with a diffusion model prior. The key idea is to sample from a chain of approximate conditional posteriors, one for each stage of the reverse diffusion process, which are obtained by the Laplace approximation. Our approximations are motivated by posterior sampling with a Gaussian prior, and inherit its simplicity and efficiency. They are asymptotically consistent and perform well empirically on a variety of contextual bandit problems.

## 1 Introduction

A *multi-armed bandit* [27, 6, 30] is an online learning problem where an agent sequentially interacts with an environment over $n$ rounds with the goal of maximizing its rewards. In each round, it takes an *action* and receives its *stochastic reward*. The mean rewards of the actions are unknown *a priori* and must be learned. This leads to the *exploration-exploitation dilemma*: *explore* actions to learn about them or *exploit* the action with the highest estimated reward. Bandits have been successfully applied to problems where uncertainty modeling and adaptation are beneficial, such recommender systems [32, 54, 25, 35] and hyper-parameter optimization [34].

Contextual bandits [29, 32] with linear [13, 1] and *generalized linear models (GLMs)* [17, 33, 2, 26] have become popular due to the their flexibility and efficiency. The features in these models can be hand-crafted or learned from historic data [40], and the models can be also updated incrementally [1, 24]. While the original algorithms for linear and GLM bandits were based on *upper confidence bounds (UCBs)* [13, 1, 17], *Thompson sampling (TS)* is more popular in practice [11, 3, 42, 44]. The key idea in TS is to explore by sampling from the posterior distribution of model parameter $\theta_*$. TS uses the prior knowledge about $\theta_*$ to speed up exploration [11, 40, 36, 9, 21, 20, 5]. When the prior is a multivariate Gaussian, the posterior of $\theta_*$ can be updated and sampled from efficiently [11]. This prior has a limited expressive power, because it cannot even represent multimodal distributions. To address this, we study posterior sampling with a diffusion prior. The main benefit of such priors is that they can represent complex distributions and be learned from data.

We make the following contributions. First, we propose novel posterior sampling approximations for linear models and GLMs with a diffusion model prior. The key idea is to sample from a chain of approximate conditional posteriors, one for each stage of the reverse process, which are estimated in a closed form. In linear models, each conditional is a product of two Gaussians, representing prior knowledge and diffused evidence (Theorem 2). In GLMs, each conditional is obtained by a Laplace approximation, which mixes prior knowledge and evidence (Theorem 4). Our approximations are motivated by posterior sampling with Gaussian priors, and inherit its simplicity and efficiency. Prior works (Section 7) sampled from the posterior using the likelihood score, and their approximations become unstable when the score is high. We combine the likelihood with conditional priors, in each

---

*The work was done at AWS AI Labs.

stage of the diffusion model, using the Laplace approximation. The resulting posterior concentrates at a single point and can be sampled from efficiently even if the likelihood score is high. We prove that this approximation is asymptotically consistent.

Our second contribution is in theory. We properly derive our posterior approximations (Theorems 2 and 4) and show their asymptotic consistency (Theorem 3). The key idea in the proof of Theorem 3 is that the conditional posteriors concentrate at a scaled unknown model parameter as the number of observations increases. While this claim is asymptotic, it is an expected property of a posterior distribution. Many prior works, such as Chung et al. [12], do not propose asymptotically consistent approximations. All of our main results rely on a novel approximation of clean samples by scaled diffused samples (Section 4.3). The most challenging part of the analysis is Theorem 3, where we analyze an asymptotic behavior of a chain of $T$ dependent random vectors.

Our last contribution is an empirical evaluation on contextual bandits. We focus on bandits because the ability to represent all levels of uncertainty precisely is critical for exploration. Our experiments show that a score-based method fails to do so (Section 6.2). Note that our posterior approximations are general and not restricted to bandits.

## 2  Setting

We start with introducing our notation. Random variables are capitalized, except for Greek letters like $\theta$. We denote the marginal and conditional probabilities under probability measure $p$ by $p(X = x)$ and $p(X = x \mid Y = y)$, respectively. When the random variables are clear from context, we write $p(x)$ and $p(x \mid y)$. We denote by $X_{n:m}$ and $x_{n:m}$ a collection of random variables and their values, respectively. For a positive integer $n$, we define $[n] = \{1, \ldots, n\}$. The indicator function is $\mathbb{1}\{\cdot\}$. The $i$-th entry of vector $v$ is $v_i$. If the vector is already indexed, such as $v_j$, we write $v_{j,i}$. We denote the maximum and minimum eigenvalues of matrix $M \in \mathbb{R}^{d \times d}$ by $\lambda_1(M)$ and $\lambda_d(M)$, respectively.

The posterior sampling problem can be formalized as follows. Let $\theta_* \in \Theta$ be an unknown *model parameter* and $\Theta \subseteq \mathbb{R}^d$ be the space of model parameters. Let $h = \{(\phi_\ell, y_\ell)\}_{\ell \in [N]}$ be the *history* of $N$ noisy observations of $\theta_*$, where $\phi_\ell \in \mathbb{R}^d$ is the feature vector for $y_\ell \in \mathbb{R}$. We assume that

$$y_\ell = g(\phi_\ell^\top \theta_*) + \varepsilon_\ell \,, \tag{1}$$

where $g : \mathbb{R} \to \mathbb{R}$ is the *mean function* and $\varepsilon_\ell$ is an independent zero-mean $\sigma^2$-sub-Gaussian noise for $\sigma > 0$. Let $p(h \mid \theta_*)$ be the *likelihood* of observations in history $h$ under model parameter $\theta_*$ and $p(\theta_*)$ be its *prior probability*. By Bayes' rule, the posterior distribution of $\theta_*$ given $h$ is

$$p(\theta_* \mid h) \propto p(h \mid \theta_*) \, p(\theta_*) \,. \tag{2}$$

We want to sample from $p(\cdot \mid h)$ efficiently when the prior distribution is represented by a diffusion model. As a stepping stone, we review existing posterior formulas for multivariate Gaussian priors. This motivates our solution for diffusion model priors.

### 2.1  Linear Model

The posterior of $\theta_*$ in linear models can be derived as follows.

**Assumption 1.** *Let $g$ in (1) be an identity and $\varepsilon_\ell \sim \mathcal{N}(0, \sigma^2)$. Then the likelihood of $h$ under model parameter $\theta_*$ is $p(h \mid \theta_*) \propto \exp[-\sum_{\ell=1}^N (y_\ell - \phi_\ell^\top \theta_*)^2 / (2\sigma^2)]$.*

Let $p(\theta_*) = \mathcal{N}(\theta_*; \theta_0, \Sigma_0)$ be the prior distribution of $\theta_*$, where $\theta_0 \in \mathbb{R}^d$ and $\Sigma_0 \in \mathbb{R}^{d \times d}$ are the prior mean and covariance, respectively. Then $p(\theta_* \mid h) \propto \mathcal{N}(\theta_*; \hat{\theta}, \hat{\Sigma})$ [10], where

$$\hat{\theta} = \hat{\Sigma} \left( \Sigma_0^{-1} \theta_0 + \sigma^{-2} \sum_{\ell=1}^N \phi_\ell y_\ell \right) \,, \quad \hat{\Sigma} = \left( \Sigma_0^{-1} + \sigma^{-2} \sum_{\ell=1}^N \phi_\ell \phi_\ell^\top \right)^{-1} \,,$$

are the posterior mean and covariance, respectively. In this work, we write them equivalently as

$$\hat{\theta} = \hat{\Sigma}(\Sigma_0^{-1} \theta_0 + \bar{\Sigma}^{-1} \bar{\theta}) \,, \quad \hat{\Sigma} = (\Sigma_0^{-1} + \bar{\Sigma}^{-1})^{-1} \,, \tag{3}$$

where $\bar{\theta} = \sigma^{-2} \bar{\Sigma} \sum_{\ell=1}^N \phi_\ell y_\ell$ and $\bar{\Sigma}^{-1} = \sigma^{-2} \sum_{\ell=1}^N \phi_\ell \phi_\ell^\top$ are the empirical mean and inverse of its covariance, respectively. Therefore, the posterior of $\theta_*$ is a product of two multivariate Gaussians: $\mathcal{N}(\theta_0, \Sigma_0)$ representing prior knowledge about $\theta_*$ and $\mathcal{N}(\bar{\theta}, \bar{\Sigma})$ representing empirical evidence.

---

**Algorithm 1** `IRLS`: Iteratively reweighted least squares.

1: **Input:** Prior parameters $\theta_0$ and $\Sigma_0$, history of observations $h = \{(\phi_\ell, y_\ell)\}_{\ell \in [N]}$

2: Initialize $\hat{\theta} \in \mathbb{R}^d$
3: **repeat**
4:     **for** stage $\ell = 1, \ldots, N$ **do**
5:         $z_\ell \leftarrow \phi_\ell^\top \hat{\theta} + (y_\ell - g(\phi_\ell^\top \hat{\theta}))/\dot{g}(\phi_\ell^\top \hat{\theta})$
6:     $\hat{\Sigma} \leftarrow \left( \Sigma_0^{-1} + \sum_{\ell=1}^{N} \dot{g}(\phi_\ell^\top \hat{\theta}) \phi_\ell \phi_\ell^\top \right)^{-1}$
7:     $\hat{\theta} \leftarrow \hat{\Sigma} \left( \Sigma_0^{-1} \theta_0 + \sum_{\ell=1}^{N} \dot{g}(\phi_\ell^\top \hat{\theta}) \phi_\ell z_\ell \right)$
8: **until** $\hat{\theta}$ converges

9: **Output:** Posterior mean $\hat{\theta}$ and covariance $\hat{\Sigma}$

---

## 2.2 Generalized Linear Model

*Generalized linear models (GLMs)* [37] extend linear models (Section 2.1) to non-linear monotone *mean functions* $g$ in (1). For instance, in logistic regression, $g(u) = 1/(1 + \exp[-u])$ is a sigmoid. The likelihood of observations in GLMs has the following form [26].

**Assumption 2.** *Let $h = \{(\phi_\ell, y_\ell)\}_{\ell \in [N]}$ be a history of $N$ observations under mean function $g$ and the corresponding noise. Then $\log p(h \mid \theta_*) \propto \sum_{\ell=1}^{N} y_\ell \phi_\ell^\top \theta_* - b(\phi_\ell^\top \theta_*) + c(y_\ell)$, where $c$ is a real function and $b$ is a function whose derivative is the mean function, $\dot{b} = g$.*

The posterior distribution of $\theta_*$ in GLMs does not have a closed form in general [10]. Therefore, it is often approximated by the *Laplace approximation*. Let the prior distribution of the model parameter be $p(\theta_*) = \mathcal{N}(\theta_*; \theta_0, \Sigma_0)$, as in Section 2.1. Then the Laplace approximation is $\mathcal{N}(\hat{\theta}, \hat{\Sigma})$, where $\hat{\theta}$ is the *maximum a posteriori (MAP) estimate* of $\theta_*$ and $\hat{\Sigma}$ is the corresponding covariance. Note that the Laplace approximation can be applied to non-Gaussian priors.

The MAP estimate $\hat{\theta}$ can be obtained by *iteratively reweighted least squares (IRLS)* [52], which we present in Algorithm 1. `IRLS` is a Newton-type algorithm that computes $\hat{\theta}$ iteratively (lines 6 and 7). It converges to the optimal solution due to the strong convexity of the problem. The solution has a similar structure to (3). That is, $\mathcal{N}(\hat{\theta}, \hat{\Sigma})$ is a product of two multivariate Gaussians, representing prior knowledge about $\theta_*$ and empirical evidence. The new quantities in GLMs are the derivative of the mean function $\dot{g}$ and pseudo-observations $z_\ell$ (line 5), which play the role of observations $y_\ell$ in Section 2.1.

## 2.3 Towards Diffusion Model Priors

The assumption that $p(\theta_*) = \mathcal{N}(\theta_*; \theta_0, \Sigma_0)$ is limiting, for instance because it precludes multimodal priors. We relax it by representing $p(\theta_*)$ by a diffusion model, which we call a *diffusion model prior*. We propose efficient posterior sampling approximations for this prior, where the prior and empirical evidence are mixed similarly to (3) and `IRLS`. We review diffusion models next.

# 3 Diffusion Models

Diffusion models [46, 19] are generative models trained by diffusing samples from unknown and hard to represent distributions. They can be viewed in multiple ways [49]. We adopt the probabilistic formulation and presentation of Ho et al. [19]. A *diffusion model* is a graphical model with $T$ stages indexed by $t \in [T]$. Each stage $t$ is associated with a *latent variable* $S_t \in \mathbb{R}^d$. A *sample* from the model is represented by an *observed variable* $S_0 \in \mathbb{R}^d$. We visualize a diffusion model in Figure 1. In the *forward process*, a clean sample $s_0$ is diffused through a sequence of variables $S_1, \ldots, S_T$. This process is used to learn the *reverse process*, where the clean sample $s_0$ is generated through a sequence of variables $S_T, \ldots, S_0$. To sample $s_0$ from the posterior (Section 4), we add a random variable $H$ that represents partial information about $s_0$. We introduce forward and reverse diffusion

Forward process (probability measure $q$)   Reverse process (probability measure $p$)
$$S_T \leftarrow S_{T-1} \leftarrow \cdots \leftarrow S_1 \leftarrow S_0 \qquad S_T \rightarrow S_{T-1} \rightarrow \cdots \rightarrow S_1 \rightarrow S_0 \rightarrow H$$

Figure 1: Graphical models of the forward and reverse processes in the diffusion model. The variable $H$ represents partial information about $S_0$.

processes next. Learning of the reverse process is described in Appendix B. While this is a critical component of diffusion models, it is not necessary to introduce our posterior approximations.

**Forward process.** In the forward process, a clean sample $s_0$ is diffused through a chain of latent variables $S_1, \ldots S_T$ (Figure 1). We denote the probability measure under this process by $q$ and define its joint probability distribution as

$$q(s_{1:T} \mid s_0) = \prod_{t=1}^{T} q(s_t \mid s_{t-1}), \quad \forall t \in [T] : q(s_t \mid s_{t-1}) = \mathcal{N}(s_t; \sqrt{\alpha_t} s_{t-1}, \beta_t I_d), \quad (4)$$

where $q(s_t \mid s_{t-1})$ is the conditional density of mapping a less diffused $s_{t-1}$ to a more diffused $s_t$. The diffusion rate is set by parameters $\alpha_t \in (0, 1)$ and $\beta_t = 1 - \alpha_t$. The forward process is sampled from as follows. First, a clean sample $s_0$ is chosen. Then $S_t \sim q(\cdot \mid s_{t-1})$ are sampled, from $t = 1$ to $t = T$.

**Reverse process.** In the reverse process, a clean sample $s_0$ is generated through a chain of variables $S_T, \ldots, S_0$ (Figure 1). We denote the probability measure under this process by $p$ and define its joint probability distribution as

$$p(s_{0:T}) = p(s_T) \prod_{t=1}^{T} p(s_{t-1} \mid s_t), \qquad (5)$$

$$p(s_T) = \mathcal{N}(s_T; \mathbf{0}_d, I_d), \quad \forall t \in [T] : p(s_{t-1} \mid s_t) = \mathcal{N}(s_{t-1}; \mu_t(s_t), \Sigma_t),$$

where $p(s_{t-1} \mid s_t)$ is the conditional density of mapping a more diffused $s_t$ to a less diffused $s_{t-1}$. The function $\mu_t$ predicts the mean of $S_{t-1} \mid s_t$ and is learned (Appendix B). As in Ho et al. [19], we keep the covariance fixed at $\Sigma_t = \tilde{\beta}_t I_d$, where $\tilde{\beta}_t = \frac{1 - \bar{\alpha}_{t-1}}{1 - \bar{\alpha}_t} \beta_t$ and $\bar{\alpha}_t = \prod_{\ell=1}^{t} \alpha_\ell$. This is known as a *stable diffusion*. We make this assumption only to simplify exposition. All our derivations in Section 4 hold when $\Sigma_t$ is learned, for instance as in Bao et al. [8].

This process is called reverse because it is learned by reversing the forward process. The reverse process is sampled from as follows. First, an initial diffused sample $S_T \sim p$ is sampled. After that, $S_{t-1} \sim p(\cdot \mid s_t)$ are sampled, from $t = T$ to $t = 1$.

## 4   Posterior Sampling

This section is organized as follows. In Section 4.1, we show how to sample from a chain of random variables conditioned on observations. In Sections 4.2 and 4.4, we specialize this to the observation models in Section 2.

### 4.1   Chain Model Posterior

Let $h = \{(\phi_\ell, y_\ell)\}_{\ell \in [N]}$ denote a *history* of $N$ observations (Section 2) and $H$ be the corresponding random variable. In this section, we assume that $h$ is fixed. The Markovian structure of the reverse process (Figure 1) implies that the joint probability distribution conditioned on $h$ factors as

$$p(s_{0:T} \mid h) = p(s_T \mid h) \prod_{t=1}^{T} p(s_{t-1} \mid s_t, h).$$

Therefore, $p(s_{0:T} \mid h)$ can be sampled from efficiently by first sampling from $p(s_T \mid h)$ and then from $T$ conditional distributions $p(s_{t-1} \mid s_t, h)$. We derive these next.

**Lemma 1.** *Let $p$ be a probability measure over the reverse process (Figure 1). Then*

$$p(s_T \mid h) \propto \int_{s_0} p(h \mid s_0) \, p(s_0 \mid s_T) \, \mathrm{d}s_0 \, p(s_T),$$

$$\forall t \in [T] \setminus \{1\} : p(s_{t-1} \mid s_t, h) \propto \int_{s_0} p(h \mid s_0) \, p(s_0 \mid s_{t-1}) \, \mathrm{d}s_0 \, p(s_{t-1} \mid s_t),$$

$$p(s_0 \mid s_1, h) \propto p(h \mid s_0) \, p(s_0 \mid s_1).$$

**Algorithm 2** `LaplaceDPS`: Laplace posterior sampling with a diffusion model prior.
___
1: **Input:** Diffusion model parameters $(\mu_t, \Sigma_t)_{t \in [T]}$, history of observations $h$

2: Initial sample $S_T \sim \mathcal{N}(\hat{\mu}_{T+1}(h), \hat{\Sigma}_{T+1}(h))$
3: **for** stage $t = T, \ldots, 1$ **do**
4:     $S_{t-1} \sim \mathcal{N}(\hat{\mu}_t(S_t, h), \hat{\Sigma}_t(h))$

5: **Output:** Posterior sample $S_0$
___

*Proof.* The claim is proved in Appendix A.1. $\qquad\qquad\qquad\qquad\qquad\qquad\qquad\qquad\square$

## 4.2 Linear Model Posterior

Now we specialize Lemma 1 to the diffusion model prior (Section 3) and linear models (Section 2.1). The prior distribution is the reverse process in (5),

$$p(s_T) = \mathcal{N}(s_T; \mathbf{0}_d, I_d), \quad \forall t \in [T] : p(s_{t-1} \mid s_t) = \mathcal{N}(s_{t-1}; \mu_t(s_t), \Sigma_t).$$

The term $p(h \mid s_0)$ is the likelihood of observations in Assumption 1. The main challenge in using the lemma is that the conditional densities of clean samples $p(s_0 \mid S_T)$ and $p(s_0 \mid s_t)$ are complex [12]. To get around this, we make an additional assumption, which is discussed in Section 4.3.

**Theorem 2.** *Let $p$ be a probability measure over the reverse process (Figure 1). Let $\bar{\theta}$ and $\bar{\Sigma}^{-1}$ be defined as in (3). Suppose that*

$$\int_{s_0} p(h \mid s_0)\, p(s_0 \mid s_t)\, \mathrm{d}s_0 \propto p(h \mid s_t/\sqrt{\bar{\alpha}_t}) \tag{6}$$

*holds for all $t \in [T]$. Then $p(s_T \mid h) \propto \mathcal{N}(s_T; \hat{\mu}_{T+1}(h), \hat{\Sigma}_{T+1}(h))$, where*

$$\hat{\mu}_{T+1}(h) = \hat{\Sigma}_{T+1}(h)(\underbrace{I_d\, \mathbf{0}_d}_{\text{Prior}} + \underbrace{\bar{\Sigma}^{-1}\bar{\theta}/\sqrt{\bar{\alpha}_T}}_{\text{Evidence}}), \quad \hat{\Sigma}_{T+1}(h) = (\underbrace{I_d}_{\text{Prior}} + \underbrace{\bar{\Sigma}^{-1}/\bar{\alpha}_T}_{\text{Evidence}})^{-1}. \tag{7}$$

*For $t \in [T]$, we have $p(s_{t-1} \mid s_t, h) \propto \mathcal{N}(s_{t-1}; \hat{\mu}_t(s_t, h), \hat{\Sigma}_t(h))$, where*

$$\hat{\mu}_t(s_t, h) = \hat{\Sigma}_t(h)(\underbrace{\Sigma_t^{-1}\mu_t(s_t)}_{\text{Prior}} + \underbrace{\bar{\Sigma}^{-1}\bar{\theta}/\sqrt{\bar{\alpha}_{t-1}}}_{\text{Evidence}}), \quad \hat{\Sigma}_t(h) = (\underbrace{\Sigma_t^{-1}}_{\text{Prior}} + \underbrace{\bar{\Sigma}^{-1}/\bar{\alpha}_{t-1}}_{\text{Evidence}})^{-1}. \tag{8}$$

*Proof.* The proof is in Appendix A.2. It has four steps. First, we fix stage $t$ and apply approximation (6). Second, we rewrite the likelihood as in (3). Third, we reparameterize it as a function of $s_t$. At the end, we combine the likelihood with the Gaussian prior using Lemma 6 in Appendix A.5. $\quad\square$

The algorithm that samples from the posterior distribution in Theorem 2 is presented in Algorithm 2. We call it *Laplace diffusion posterior sampling (`LaplaceDPS`)* because its generalization to GLMs uses the Laplace approximation. `LaplaceDPS` samples from a chain of products of two distributions: one distribution represents the pre-trained diffusion model and does not depend on history $h$, and the other represents the history $h$. The sampling is implemented as follows. The initial variable $S_T$ is sampled conditioned on $h$ (line 2) from the distribution in (7). This distribution is a product of the $h$-independent prior $\mathcal{N}(\mathbf{0}_d, I_d)$ and the $h$-dependent distribution of the diffused evidence up to stage $T$, $\mathcal{N}(\sqrt{\bar{\alpha}_T}\bar{\theta}, \bar{\alpha}_T\bar{\Sigma})$. Then, for any $t \in [T]$, $S_{t-1}$ is sampled conditioned on $s_t$ and evidence $h$ (line 4) from the distribution in (8). This distribution is a product of the $h$-independent conditional prior $\mathcal{N}(\mu_t(s_t), \Sigma_t)$, from the pre-trained model, and the $h$-dependent distribution of the diffused evidence up to stage $t-1$, $\mathcal{N}(\sqrt{\bar{\alpha}_{t-1}}\bar{\theta}, \bar{\alpha}_{t-1}\bar{\Sigma})$. The last variable $S_0$ is the clean sample. When compared to Section 2, the prior and evidence are mixed conditionally in a $T$-stage chain. This increases the computational cost $T$ times, as discussed in Section 8.

## 4.3 Key Approximation in Theorem 2

Now we motivate our assumption in (6). Simply put, we assume that $s_0 = s_t/\sqrt{\bar{\alpha}_t}$, where $s_0$ is a clean sample and $s_t$ is the corresponding diffused sample in stage $t$. This is motivated by the forward process, which relates $s_t$ and $s_0$ as $s_t = \sqrt{\bar{\alpha}_t}s_0 + \sqrt{1 - \bar{\alpha}_t}\varepsilon_t$, where $\varepsilon_t \sim \mathcal{N}(\mathbf{0}_d, I_d)$ is a standard

---
**Algorithm 3** Contextual Thompson sampling.
---
1: **for** round $k = 1, \ldots, n$ **do**
2:      Sample $\tilde{\theta}_k \sim p(\cdot \mid h_k)$, where $p(\cdot \mid h_k)$ is the posterior distribution in (2)
3:      Take action $a_k \leftarrow \arg\max_{a \in \mathcal{A}} r(x_k, a; \tilde{\theta}_k)$ and observe reward $y_k$
---

Gaussian noise [19]. After rearranging, we get $s_0 = (s_t - \sqrt{1 - \bar{\alpha}_t}\varepsilon_t)/\sqrt{\bar{\alpha}_t}$, and therefore $s_0$ can be viewed as a random variable with mean $s_t/\sqrt{\bar{\alpha}_t}$. The consequence of (6) is that the likelihood becomes a function of $s_t$, which yields a closed form when multiplied by the conditional prior, also a function of $s_t$. Our approximation can be also viewed as the Tweedie's formula in Chung et al. [12] where the score component is neglected.

Our approximation has several notable properties. First, $\sqrt{(1 - \bar{\alpha}_t)/\bar{\alpha}_t} \to 0$ as $t \to 1$. Therefore, it becomes more precise in later stages of the reverse process. Second, in the absence of evidence $h$, the approximation vanishes, and all posterior distributions in Theorem 2 reduce to the priors in (5). Finally, as the number of observations increases, sampling from the posterior in Theorem 2 is asymptotically consistent.

**Theorem 3.** *Fix $\theta_* \in \mathbb{R}^d$. Let $\tilde{\theta} \leftarrow \texttt{LaplaceDPS}((\mu_t, \Sigma_t)_{t \in [T]}, h)$, where $h = \{(\phi_\ell, y_\ell)\}_{\ell \in [N]}$ is a history of $N$ observations. Suppose that $\lambda_d(\bar{\Sigma}^{-1}) \to \infty$ as $N \to \infty$, where $\bar{\Sigma}$ is defined in (3). Then $\mathbb{P}\left(\lim_{N \to \infty} \|\tilde{\theta} - \theta_*\|_2 = 0\right) = 1$.*

*Proof.* The proof is in Appendix A.3. The key idea is that the conditional posteriors in (7) and (8) concentrate at a scaled unknown model parameter $\theta_*$ as the number of observations increases, which we formalize as $\lambda_d(\bar{\Sigma}^{-1}) \to \infty$.     □

The bound in Theorem 3 can be interpreted as follows. The sampled parameter $\tilde{\theta}$ approaches the true unknown parameter $\theta_*$ as the number of observations $N$ increases. To guarantee that the posterior shrinks uniformly in all directions, we assume that the number of observations in all directions grows linearly with $N$. This is akin to assuming that $\lambda_d(\bar{\Sigma}^{-1}) = \Omega(N)$. This lower bound can be attained in linear models by getting observations according to the D-optimal design [39].

## 4.4 GLM Posterior

The Laplace approximation in GLMs (Section 2.2) naturally generalizes the exact posterior distribution in linear models (Section 2.1). We generalize Theorem 2 to GLMs along the same lines.

**Theorem 4.** *Let $p$ be a probability measure over the reverse process (Figure 1). Suppose that (6) holds for all $t \in [T]$. Then $p(s_T \mid h) \propto \mathcal{N}(s_T; \hat{\mu}_{T+1}(h), \hat{\Sigma}_{T+1}(h))$, where*

$$\hat{\mu}_{T+1}(h) = \sqrt{\bar{\alpha}_T}\dot{\theta}_{T+1}, \quad \hat{\Sigma}_{T+1}(h) = \bar{\alpha}_T \dot{\Sigma}_{T+1}, \quad \dot{\theta}_{T+1}, \dot{\Sigma}_{T+1} \leftarrow \texttt{IRLS}(\mathbf{0}_d, I_d/\bar{\alpha}_T, h).$$

*For $t \in [T]$, we have $p(s_{t-1} \mid s_t, h) \propto \mathcal{N}(s_{t-1}; \hat{\mu}_t(s_t, h), \hat{\Sigma}_t(h))$, where*

$$\hat{\mu}_t(s_t, h) = \sqrt{\bar{\alpha}_{t-1}}\dot{\theta}_t, \quad \hat{\Sigma}_t(h) = \bar{\alpha}_{t-1}\dot{\Sigma}_t, \quad \dot{\theta}_t, \dot{\Sigma}_t \leftarrow \texttt{IRLS}(\mu_t(s_t)/\sqrt{\bar{\alpha}_{t-1}}, \Sigma_t/\bar{\alpha}_{t-1}, h).$$

*Proof.* The proof is in Appendix A.4. It has four steps. First, we fix stage $t$ and apply approximation (6). Second, we reparameterize the prior, from a function of $s_t$ to a function of $s_t/\sqrt{\bar{\alpha}_t}$. Third, we combine the likelihood with the prior using the Laplace approximation. Finally, we repameterize the posterior, from a function of $s_t/\sqrt{\bar{\alpha}_t}$ to a function of $s_t$.     □

Similarly to Theorem 2, the distributions in Theorem 4 mix evidence with the diffusion model prior. However, this is done implicitly in $\texttt{IRLS}$. The posterior can be sampled from using $\texttt{LaplaceDPS}$, where the mean and covariances would be taken from Theorem 4. Note that Theorem 2 is a special case of Theorem 4 where the mean function $g$ is an identity.

# 5 Application to Contextual Bandits

Now we apply our posterior sampling approximations (Section 4) to contextual bandits. A *contextual bandit* [29, 32] is a classic model for sequential decision making under uncertainty where the agent takes actions conditioned on context. We denote the *action set* by $\mathcal{A}$ and the *context set* by $\mathcal{X}$. The *mean reward* for taking action $a \in \mathcal{A}$ in context $x \in \mathcal{X}$ is $r(x, a; \theta_*)$, where $r : \mathcal{X} \times \mathcal{A} \times \Theta \to \mathbb{R}$ is a *reward function* and $\theta_* \in \Theta$ is a *model parameter* (Section 2). The agent interacts with the bandit for $n$ rounds indexed by $k \in [n]$. In round $k$, it observes a *context* $x_k \in \mathcal{X}$, takes an *action* $a_k \in \mathcal{A}$, and observes its *stochastic reward* $y_k = r(x_k, a_k; \theta_*) + \varepsilon_k$ with independent noise $\varepsilon_k$. We assume that the noise is zero-mean $\sigma^2$-sub-Gaussian for $\sigma > 0$. The objective of the agent is to maximize its cumulative reward in $n$ rounds, or equivalently to minimize its cumulative regret. We define the *n-round regret* as

$$R(n) = \sum_{k=1}^n \mathbb{E}\left[ r(x_k, a_{k,*}; \theta_*) - r(x_k, a_k; \theta_*) \right], \tag{9}$$

where $a_{k,*} = \arg\max_{a \in \mathcal{A}} r(x_k, a; \theta_*)$ is the optimal action in round $k$.

Arguably the most popular method for solving contextual bandit problems is Thompson sampling [50, 11, 3]. The key idea in TS is to use the posterior distribution of $\theta_*$ to explore. This is done as follows. In round $k$, the model parameter is drawn from the posterior in (2), $\tilde{\theta}_k \sim p(\cdot \mid h_k)$, where $h_k$ is the *history* of all interactions up to round $k$. After that, the agent takes the action with the highest mean reward under $\tilde{\theta}_k$. The pseudo-code of this algorithm is given in Algorithm 3.

A *linear bandit* [13] has a linear reward function $r(x, a; \theta_*) = \phi(x, a)^\top \theta_*$, where $\phi : \mathcal{X} \times \mathcal{A} \to \mathbb{R}^d$ is a *feature extractor*. The feature extractor can be non-linear in $x$ and $a$. Therefore, linear bandits can be applied to non-linear functions of $x$ and $a$. The feature extractor can be either learned [40] or hand-crafted. We denote the feature vector of the action in round $k$ by $\phi_k = \phi(x_k, a_k)$. Therefore, the *history* of interactions up to round $k$ is $h_k = \{(\phi_\ell, y_\ell)\}_{\ell \in [k-1]}$. When the prior distribution is a Gaussian, $p(\theta_*) = \mathcal{N}(\theta_*; \theta_0, \Sigma_0)$, the posterior in round $k$ is a Gaussian in (3) for $h = h_k$. When the prior is a diffusion model, we propose sampling from the posterior using

$$\tilde{\theta}_k \leftarrow \texttt{LaplaceDPS}((\mu_t, \Sigma_t)_{t \in [T]}, h_k), \tag{10}$$

where $\hat{\mu}_t$ and $\hat{\Sigma}_t$ in `LaplaceDPS` are computed as in Theorem 2. We call this algorithm `DiffTS`.

A *generalized linear bandit* [17, 24, 33, 26] is an extension of linear bandits to generalized linear models (Section 2.2). When $p(\theta_*) = \mathcal{N}(\theta_*; \theta_0, \Sigma_0)$, the Laplace approximation to the posterior is a Gaussian (Section 2.2). When the prior is a diffusion model, we propose posterior sampling using (10), where $\hat{\mu}_t$ and $\hat{\Sigma}_t$ in `LaplaceDPS` are computed as in Theorem 4.

# 6 Experiments

We conduct three experiments: synthetic problems in 2 dimensions (Section 6.2 and Appendix C.1), a recommender system (Section 6.3), and a classification problem (Appendix C.2). In addition, we conduct an ablation study in Appendix C.3, where we vary the number of training samples for the diffusion prior and the number of diffusion stages $T$.

## 6.1 Experimental Setup

We have four baselines. Three baselines are variants of contextual Thompson sampling [11, 3]: with an uninformative Gaussian prior (`TS`), a learned Gaussian prior (`TunedTS`), and a learned Gaussian mixture prior (`MixTS`) [22]. The last baseline is diffusion posterior sampling (`DPS`) of Chung et al. [12]. We implement all TS baselines as described in Section 5. The uninformative prior is $\mathcal{N}(\mathbf{0}_d, I_d)$. `MixTS` is used only in linear bandit experiments because the logistic regression variant does not exist. The TS baselines are chosen to cover various levels of prior information. Our implementation of `DPS` is described in Appendix D. We also experimented with frequentist baselines, such as `LinUCB` [1] and the $\varepsilon$-greedy policy. They performed worse than `TS` and thus we do not report them.

Each experiment is set up as follows. First, the prior distribution of $\theta_*$ is specified: it can be synthetic or estimated from real-world data. Second, we learn this distribution from $10\,000$ samples from it. In `DiffTS` and `DPS`, we follow Appendix B. The number of stages is $T = 100$ and the diffusion factor

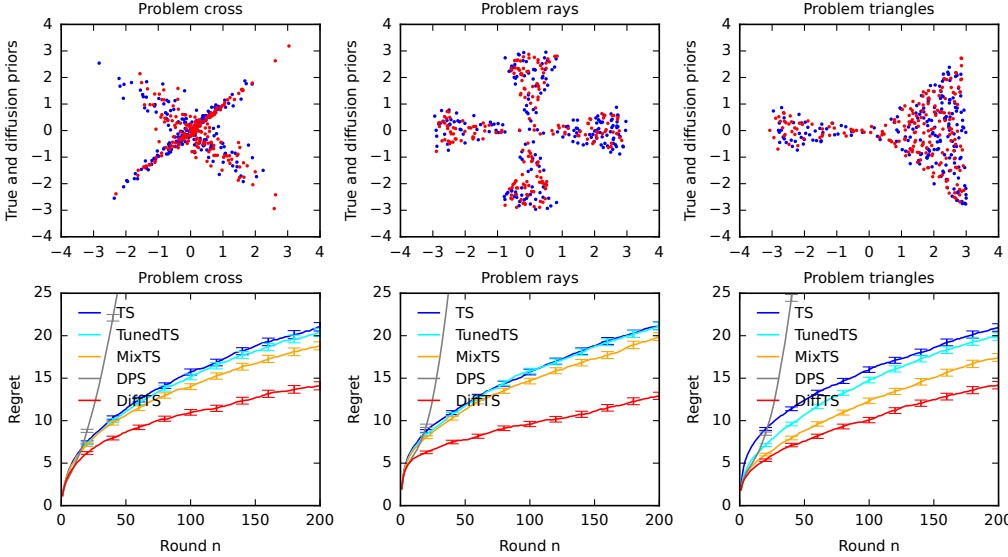

Figure 2: Evaluation of `DiffTS` on three synthetic problems. The first row shows samples from the true (blue) and diffusion model (red) priors. The second row shows the regret of `DiffTS` and the baselines as a function of round $n$.

is $\alpha_t = 0.97$. Since $0.97^{100} \approx 0.05$, most of the information in the training samples is diffused. The regressor in Appendix B is a 2-layer neural network with ReLU activations. In `TunedTS`, we fit the mean and covariance using maximum likelihood estimation. In `MixTS`, we fit the Gaussian mixture using SCIKIT-LEARN. All algorithms are evaluated on $\theta_*$ sampled from the true prior. The regret is computed as defined in (9). All error bars are standard errors of the estimates.

## 6.2 Synthetic Experiment

The first experiment is on three synthetic problems. Each problem is a linear bandit (Section 5) with $K = 100$ actions in $d = 2$ dimensions. The reward noise is $\sigma = 1$. The feature vectors of actions are sampled uniformly at random from a unit ball. The prior distributions of $\theta_*$ are shown in Figure 2. The first is a mixture of two Gaussians and the last can be approximated well by a mixture of two Gaussians. We implement `MixTS` with two mixture components. Therefore, it can represent the first prior exactly and approximate the last one well.

Our results are reported in Figure 2. We observe two main trends. First, samples from the diffusion prior closely resemble those from the true prior. In such cases, `DiffTS` is expected to perform well and even outperforms `MixTS`, because it has a better representation of the prior. We observe this in all problems. Second, `DPS` diverges as the number of rounds increases. This is because `DPS` uses an approximation based on the likelihood score (Section 7), which is unstable when the score is high. This happens despite our best efforts to tune `DPS` (Appendix D). We report results on another three synthetic problems in Appendix C.1.

`DiffTS` should be $T$ times more computationally costly than `TS` with a Gaussian prior (Section 4.2). We observe this empirically. As an example, the average cost of 100 runs of `DiffTS` on any problem in Figure 2 is 12 seconds. The average cost of `TS` is 0.1 seconds. The computation and accuracy can be traded off, and we investigate this in Appendix C.3. In the cross problem, we vary the number of diffusion stages from $T = 1$ to $T = 300$. We observe that the computational cost is linear in $T$ and the regret drops quickly from 26 at $T = 1$ to 15 at $T = 50$.

## 6.3 MovieLens Experiment

In the second experiment, we learn to recommend an item to randomly arriving users. The problem is simulated using the MovieLens 1M dataset [28], with one million ratings for 3 706 movies from 6 040 users. We subtract the mean rating from all ratings and complete the sparse rating matrix $M$ by

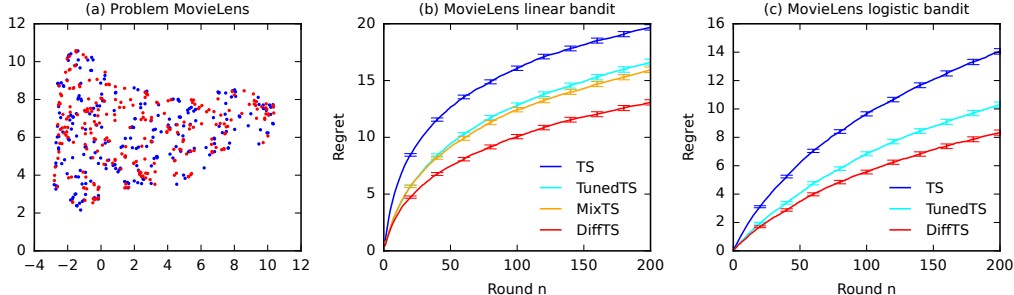

Figure 3: Evaluation of `DiffTS` on the MovieLens dataset: (a) shows samples from the true (blue) and diffusion model (red) priors, (b) shows regret in the linear bandit, and (c) shows regret in the logistic bandit.

alternating least squares [14] with rank $d = 5$. The learned factorization is $M = UV^\top$. The $i$-th row of $U$, denoted by $U_i$, represents user $i$. The $j$-th row of $V$, denoted by $V_j$, represents movie $j$. We use movie embeddings $V_j$ as model parameters and user embeddings $U_i$ as features of the actions. The movies are items.

We experiment with both linear and logistic bandits. In both, an item is initially chosen randomly from $V_j$ and $K = 10$ actions are chosen randomly from $U_i$ in each round. In the linear bandit, the mean reward of item $j$ for user $i$ is $U_i^\top V_j$. The reward noise is $\sigma = 0.75$, and we estimate it from data. In the logistic bandit, the mean reward is $g(U_i^\top V_j)$, where $g$ is a sigmoid.

Our MovieLens results are reported in Figure 3 and we observe similar trends to Section 6.2. First, samples from the diffusion prior closely resemble those from the true prior (Figure 3a). Since the problem is higher dimensional, we visualize the overlap using UMAP [45]. Second, `DiffTS` has a lower regret than all baselines, in both linear (Figure 3b) and logistic (Figure 3c) bandits. Finally, `MixTS` barely outperforms `TunedTS`. We observe this trend consistently in higher dimensions, and this motivated our work on online learning with more complex priors.

## 7 Related Work

We start with reviewing related works on bandits with diffusion models. Hsieh et al. [23] proposed Thompson sampling with a diffusion model prior for $K$-armed bandits. There are multiple technical differences from our work. First, the diffusion model in Hsieh et al. [23] is over scalars representing individual arms. Our model is over vectors representing model parameters, and thus can be applied to contextual bandits. Second, the approximations are different. In stage $t$, Hsieh et al. [23] sample from two distributions: the conditional prior and the distribution of the diffused empirical mean up to stage $t$. Then they take a weighted sum of the samples. We sample only once, from the posterior distribution that combines the conditional prior in stage $t$ and likelihood. Therefore, the method of Hsieh et al. [23] can be viewed as a non-contextual variant of our method, where posterior sampling is done by weighting samples from the prior and empirical distributions. Finally, Hsieh et al. [23] do not analyze their approximation.

Aouali [4] proposed and analyzed contextual bandits with a linear diffusion model prior: $\mu_t(s_t)$ in (5) is linear in $s_t$ and $q(s_0)$ is a Gaussian. Therefore, this model is a linear Gaussian model and not a general diffusion model, as in our work.

The closest related work on posterior sampling in diffusion models is `DPS` of Chung et al. [12]. The key idea in `DPS` is to sample from the posterior distribution using the likelihood score $\nabla \log p(h \mid \theta)$, where $p(h \mid \theta)$ is the likelihood (Assumptions 1 and 2). Note that $\nabla \log p(h \mid \theta)$ grows linearly in $N$ because the history $h$ in $p(h \mid \theta)$ involves $N$ terms. Therefore, `DPS` becomes unstable as $N \to \infty$. We show it empirically in Section 6.2 and discuss the implementation of `DPS` in Appendix D, which was tuned to improve its stability.

Many other posterior sampling methods for diffusion models have been proposed recently: a sequential Monte Carlo approximation for the conditional reverse process [53], a variant of `DPS` with an uninformative prior [38], a pseudo-inverse approximation to the likelihood of evidence [48], and

posterior sampling in latent diffusion models [41]. All of these methods rely on the likelihood score $\nabla \log p(h \mid \theta)$ and thus become unstable as the number of observations $N$ increases. Our posterior approximations do not have this issue because they are based on the product of prior and evidence distributions (Theorems 2 and 4), and thus gradient-free. They work well across different levels of uncertainty (Section 6) and do not require tuning.

We note that posterior sampling is a special form of inference-time guidance in diffusion models. Other approaches are conditional pre-training [15], a constraint in the reverse process [18], refining the null-space content [51], solving an optimization problem that pushes the reverse process towards evidence [47], and aligning the reverse process with the prompt [7].

## 8 Conclusions

We propose posterior sampling approximations for diffusion models priors. These approximations are contextual, and can be implemented efficiently in linear models and GLMs. We analyze them and evaluate them empirically on contextual bandit problems. Our method has two main limitations.

**Computational cost.** The cost of posterior sampling in `LaplaceDPS` with $T$ stages is about $T$ times higher than that of posterior sampling with a Gaussian prior (Section 2). We validate it empirically in Section 6.2. We plot the sampling time as a function of $T$ in Figure 6c (Appendix C.3).

**Learning cost and hyper-parameter tuning.** In all experiments, the number of diffusion stages is $T = 100$ and the diffusion rate is set such that most of the signal diffuses. The regressor is a 2-layer neural network and we learn it from $10\,000$ samples from the prior. These settings resulted in stable performance in all our experiments (Section 6). However, they clearly impact the performance. We plot the regret as a function of the number of training samples in Figure 6a and as a function of $T$ in Figure 6b. When $T$ or the number of training samples is small, `DiffTS` performs very similarly to posterior sampling with a Gaussian prior. In summary, there is no benefit in these cases.

**Future work.** We develop novel posterior approximations rather than bounding their regret. This is because the existing approximations are unstable and may diverge in the online setting (Sections 6.2 and 7). We believe that a proper regret analysis of `DiffTS` is possible and would require bounding two errors. The first error arises because the reverse process does not reverse the forward process exactly (Appendix B). The second error arises because our posterior distributions are approximate (Section 4.3). One possibility is to start with prior works that already showed the utility of complex priors. For instance, Russo and Van Roy [43] proved a $O(\sqrt{\Gamma H(A_*)n})$ regret bound for a linear bandit, where $\Gamma$ is the maximum ratio of regret to information gain and $H(A_*)$ is the entropy of the distribution of the optimal action under the prior. This bound holds for any prior and says that a lower entropy $H(A_*)$, which corresponds to more informative priors, yields a lower regret.

We also believe that our ideas can be extended beyond GLMs. The key idea in Section 4.4 is to use the Laplace approximation of the likelihood. This approximation can be computed exactly in GLMs. More generally though, it is a good approximation whenever the likelihood can be approximated well by a single Gaussian distribution. By the central limit theorem, under appropriate assumptions, this is expected for any observation model when the number of observations is large.

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

# A Proofs and Supporting Lemmas

This section contains proofs of our main claims and supporting lemmas.

## A.1 Proof of Lemma 1

All derivations are based on basic rules of probability and the chain structure in Figure 1, and are exact. From Figure 1, the joint probability distribution conditioned on $H = h$ factors as

$$p(s_{0:T} \mid h) = p(s_T \mid h) \prod_{t=1}^{T} p(s_{t-1} \mid s_{t:T}, h) = p(s_T \mid h) \prod_{t=1}^{T} p(s_{t-1} \mid s_t, h).$$

We use that $p(s_{t-1} \mid s_{t:T}, h) = p(s_{t-1} \mid s_t, h)$ in the last equality. We consider two cases.

**Derivation of** $p(s_{t-1} \mid s_t, h)$. By Bayes' rule, we get

$$p(s_{t-1} \mid s_t, h) = \frac{p(h \mid s_{t-1}, s_t) \, p(s_{t-1} \mid s_t)}{p(h \mid s_t)} \propto p(h \mid s_{t-1}) \, p(s_{t-1} \mid s_t).$$

In the last step, we use that $p(h \mid s_t)$ is a constant, since $s_t$ and $h$ are fixed, and that $p(h \mid s_{t-1}, s_t) = p(h \mid s_{t-1})$. Note that the last term $p(s_{t-1} \mid s_t)$ is the conditional prior distribution. When $t > 1$, the first term can be expressed as

$$p(h \mid s_{t-1}) = \int_{s_0} p(h, s_0 \mid s_{t-1}) \, \mathrm{d}s_0 = \int_{s_0} p(h \mid s_0, s_{t-1}) \, p(s_0 \mid s_{t-1}) \, \mathrm{d}s_0$$

$$= \int_{s_0} p(h \mid s_0) \, p(s_0 \mid s_{t-1}) \, \mathrm{d}s_0 \, .$$

In the last equality, we use that our graphical model is a chain (Figure 1), and thus $p(h \mid s_0, s_{t-1}) = p(h \mid s_0)$. Finally, we chain all identities and get that

$$p(s_{t-1} \mid s_t, h) \propto \int_{s_0} p(h \mid s_0) \, p(s_0 \mid s_{t-1}) \, \mathrm{d}s_0 \, p(s_{t-1} \mid s_t) \, . \tag{11}$$

**Derivation of** $p(s_T \mid h)$. By Bayes' rule, we get

$$p(s_T \mid h) = \frac{p(h \mid s_T) \, p(s_T)}{p(h)} \propto p(h \mid s_T) \, p(s_T) \, .$$

In the last step, we use that $p(h)$ is a constant, since $h$ is fixed. The first term can be rewritten as

$$p(h \mid s_T) = \int_{s_0} p(h, s_0 \mid s_T) \, \mathrm{d}s_0 = \int_{s_0} p(h \mid s_0, s_T) \, p(s_0 \mid s_T) \, \mathrm{d}s_0$$

$$= \int_{s_0} p(h \mid s_0) \, p(s_0 \mid s_T) \, \mathrm{d}s_0 \, .$$

Finally, we chain all identities and get that

$$p(s_T \mid h) \propto \int_{s_0} p(h \mid s_0) \, p(s_0 \mid s_T) \, \mathrm{d}s_0 \, p(s_T) \, . \tag{12}$$

This completes the derivations.

## A.2 Proof of Theorem 2

This proof has two parts.

**Derivation of** $p(s_{t-1} \mid s_t, h)$. From (6) and Assumption 1, it follows that

$$\int_{s_0} p(h \mid s_0) \, p(s_0 \mid s_{t-1}) \, \mathrm{d}s_0 \propto p(h \mid s_{t-1}/\sqrt{\bar{\alpha}_{t-1}}) \propto \mathcal{N}(s_{t-1}/\sqrt{\bar{\alpha}_{t-1}}; \bar{\theta}, \bar{\Sigma})$$

$$\propto \mathcal{N}(s_{t-1}; \sqrt{\bar{\alpha}_{t-1}}\bar{\theta}, \bar{\alpha}_{t-1}\bar{\Sigma}) \, .$$

The last step treats $\bar{\alpha}_{t-1}$ and $\bar{\Sigma}$ as constants, because the forward process and evidence $h$ are fixed. Now we apply Lemma 6 to distributions

$$p(s_{t-1} \mid s_t) = \mathcal{N}(s_{t-1}; \mu_t(s_t), \Sigma_t), \quad \mathcal{N}(s_{t-1}; \sqrt{\bar{\alpha}_{t-1}}\theta, \bar{\alpha}_{t-1}\bar{\Sigma}),$$

and get that

$$p(s_{t-1} \mid s_t, h) \propto \mathcal{N}(s_{t-1}; \hat{\mu}_t(s_t, h), \hat{\Sigma}_t(h)),$$

where $\hat{\mu}_t(s_t, h)$ and $\hat{\Sigma}_t(h)$ are defined in the claim. This is a product of two Gaussians: the prior with mean $\mu_t(s_t)$ and covariance $\Sigma_t$, and the evidence with mean $\sqrt{\bar{\alpha}_{t-1}}\theta$ and covariance $\bar{\alpha}_{t-1}\bar{\Sigma}$.

**Derivation of $p(s_T \mid h)$.** Analogously to the derivation of $p(s_{t-1} \mid s_t, h)$, we establish that

$$\int_{s_0} p(h \mid s_0)\, p(s_0 \mid s_T)\, \mathrm{d}s_0 \propto \mathcal{N}(s_T; \sqrt{\bar{\alpha}_T}\theta, \bar{\alpha}_T\bar{\Sigma}).$$

Then we apply Lemma 6 to distributions

$$p(s_T) = \mathcal{N}(s_T; \mathbf{0}_d, I_d), \quad \mathcal{N}(s_T; \sqrt{\bar{\alpha}_T}\theta, \bar{\alpha}_T\bar{\Sigma}),$$

and get that

$$p(s_T \mid h) \propto \mathcal{N}(s_T; \hat{\mu}_{T+1}(h), \hat{\Sigma}_{T+1}(h)),$$

where $\hat{\mu}_{T+1}(h)$ and $\hat{\Sigma}_{T+1}(h)$ are defined in the claim. This is a product of two Gaussians: the prior with mean $\mathbf{0}_d$ and covariance $I_d$, and the evidence with mean $\sqrt{\bar{\alpha}_T}\theta$ and covariance $\bar{\alpha}_T\bar{\Sigma}$.

### A.3 Proof of Theorem 3

We start with the triangle inequality

$$\|\tilde{\theta} - \theta_*\|_2 = \|\tilde{\theta} - \bar{\theta} + \bar{\theta} - \theta_*\|_2 \leq \|\tilde{\theta} - \bar{\theta}\|_2 + \|\bar{\theta} - \theta_*\|_2,$$

where we introduce $\bar{\theta}$ from Section 2.1. Now we bound each term on the right-hand side.

**Upper bound on $\|\tilde{\theta} - \bar{\theta}\|_2$.** This part of the proof is based on analyzing the asymptotic behavior of the conditional densities in Theorem 2.

As a first step, note that $S_T \sim \mathcal{N}(\hat{\mu}_{T+1}(h), \hat{\Sigma}_{T+1}(h))$, where

$$\hat{\mu}_{T+1}(h) = \hat{\Sigma}_{T+1}(h)(I_d\, \mathbf{0}_d + \bar{\Sigma}^{-1}\theta/\sqrt{\bar{\alpha}_T}), \quad \hat{\Sigma}_{T+1}(h) = (I_d + \bar{\Sigma}^{-1}/\bar{\alpha}_T)^{-1}.$$

Since $\lambda_d(\bar{\Sigma}^{-1}) \to \infty$, we get

$$\hat{\Sigma}_{T+1}(h) \to \bar{\alpha}_T\bar{\Sigma}, \quad \hat{\mu}_{T+1}(h) \to \sqrt{\bar{\alpha}_T}\theta.$$

Moreover, $\lambda_d(\bar{\Sigma}^{-1}) \to \infty$ implies $\lambda_1(\bar{\Sigma}) \to 0$, and thus $\lim_{N \to \infty} \|S_T - \sqrt{\bar{\alpha}_T}\theta\|_2 = 0$.

The same argument can be applied inductively to later stages of the reverse process. Specifically, for any $t \in [T]$, $S_{t-1} \sim \mathcal{N}(\hat{\mu}_t(S_t, h), \hat{\Sigma}_t(h))$, where

$$\hat{\mu}_t(S_t, h) = \hat{\Sigma}_t(h)(\Sigma_t^{-1}\mu_t(S_t) + \bar{\Sigma}^{-1}\theta/\sqrt{\bar{\alpha}_{t-1}}), \quad \hat{\Sigma}_t(h) = (\Sigma_t^{-1} + \bar{\Sigma}^{-1}/\bar{\alpha}_{t-1})^{-1}.$$

Since $\lambda_d(\bar{\Sigma}^{-1}) \to \infty$ and $S_t \to \sqrt{\bar{\alpha}_t}\theta$ by induction, we get

$$\hat{\Sigma}_t(h) \to \bar{\alpha}_{t-1}\bar{\Sigma}, \quad \hat{\mu}_t(S_t, h) \to \sqrt{\bar{\alpha}_{t-1}}\theta.$$

Moreover, $\lambda_d(\bar{\Sigma}^{-1}) \to \infty$ implies $\lambda_1(\bar{\Sigma}) \to 0$, and thus $\lim_{N \to \infty} \|S_{t-1} - \sqrt{\bar{\alpha}_{t-1}}\theta\|_2 = 0$ for any $t \in [T]$. In the last stage, $t = 1$, $\bar{\alpha}_0 = 1$, and $S_0 = \tilde{\theta}$. Therefore,

$$\lim_{N \to \infty} \|S_{t-1} - \sqrt{\bar{\alpha}_{t-1}}\theta\|_2 = \lim_{N \to \infty} \|\tilde{\theta} - \bar{\theta}\|_2 = 0.$$

**Upper bound on $\|\bar{\theta} - \theta_*\|_2$.** This part of the proof uses the definition of $\bar{\theta}$ in Section 2.1 and that $\varepsilon_\ell \sim \mathcal{N}(0, \sigma^2)$ is independent noise. By definition,

$$\bar{\theta} - \theta_* = \sigma^{-2}\bar{\Sigma}\sum_{\ell=1}^{N}\phi_\ell y_\ell - \theta_* = \sigma^{-2}\bar{\Sigma}\sum_{\ell=1}^{N}\phi_\ell(\phi_\ell^\top\theta_* + \varepsilon_\ell) - \theta_* = \sigma^{-2}\bar{\Sigma}\sum_{\ell=1}^{N}\phi_\ell\varepsilon_\ell.$$

Since $\varepsilon_\ell$ is independent zero-mean Gaussian noise with variance $\sigma^2$, $\bar\theta - \theta_*$ is a Gaussian random variable with mean $\mathbf{0}_d$ and covariance

$$\mathrm{cov}\left[\sigma^{-2}\bar\Sigma\sum_{\ell=1}^N \phi_\ell\varepsilon_\ell\right] = \sigma^{-4}\bar\Sigma\left(\sum_{\ell=1}^N \phi_\ell\mathrm{var}\left[\varepsilon_\ell\right]\phi_\ell^\top\right)\bar\Sigma = \bar\Sigma\frac{\sum_{\ell=1}^N \phi_\ell\phi_\ell^\top}{\sigma^2}\bar\Sigma = \bar\Sigma\,.$$

Since $\lambda_d(\bar\Sigma^{-1}) \to \infty$ implies $\lambda_1(\bar\Sigma) \to 0$, we get

$$\lim_{N\to\infty}\|\bar\theta - \theta_*\|_2 = 0\,.$$

This completes the proof.

### A.4  Proof of Theorem 4

This proof has two parts.

**Derivation of $p(s_{t-1} \mid s_t, h)$.** From (6), we have

$$\int_{s_0} p(h \mid s_0)\,p(s_0 \mid s_{t-1})\,\mathrm{d}s_0 \propto p(h \mid s_{t-1}/\sqrt{\bar\alpha_{t-1}})\,.$$

Since $p(s_{t-1} \mid s_t)$ is a Gaussian, we have

$$p(s_{t-1} \mid s_t) = \mathcal{N}(s_{t-1}; \mu_t(s_t), \Sigma_t) \propto \mathcal{N}(\gamma s_{t-1}; \gamma\mu_t(s_t), \gamma^2\Sigma_t)$$

for $\gamma = 1/\sqrt{\bar\alpha_{t-1}}$. Then by the Laplace approximation,

$$p(h \mid \gamma s_{t-1})\,\mathcal{N}(\gamma s_{t-1}; \gamma\mu_t(s_t), \gamma^2\Sigma_t) \propto \mathcal{N}(\gamma s_{t-1}; \dot\theta_t, \dot\Sigma_t) \propto \mathcal{N}(s_{t-1}; \dot\theta_t/\gamma, \dot\Sigma_t/\gamma^2)\,,$$

where $\dot\theta_t, \dot\Sigma_t \leftarrow \mathtt{IRLS}(\gamma\mu_t(s_t), \gamma^2\Sigma_t, h)$.

**Derivation of $p(s_T \mid h)$.** Analogously to the derivation of $p(s_{t-1} \mid s_t, h)$, we establish that

$$\int_{s_0} p(h \mid s_0)\,p(s_0 \mid s_T)\,\mathrm{d}s_0 \propto p(h \mid s_T/\sqrt{\bar\alpha_T})\,.$$

Then by the Laplace approximation for $\gamma = 1/\sqrt{\bar\alpha_T}$, we get

$$p(h \mid \gamma s_T)\,\mathcal{N}(s_T; \mathbf{0}_d, I_d) \propto \mathcal{N}(s_T; \dot\theta_{T+1}/\gamma, \dot\Sigma_{T+1}/\gamma^2)\,,$$

where $\dot\theta_{T+1}, \dot\Sigma_{T+1} \leftarrow \mathtt{IRLS}(\mathbf{0}_d, \gamma^2 I_d, h)$.

### A.5  Supporting Lemmas

We state and prove our supplementary lemmas next.

**Lemma 5.** *Let $p(x) = \mathcal{N}(x; \mu_1, \Sigma_1)$ and $q(x) = \mathcal{N}(x; \mu_2, \Sigma_2)$, where $\mu_1, \mu_2 \in \mathbb{R}^d$ and $\Sigma_1, \Sigma_2 \in \mathbb{R}^{d\times d}$. Then*

$$d(p, q) = \frac{1}{2}\left((\mu_2 - \mu_1)^\top\Sigma_2^{-1}(\mu_2 - \mu_1) + \mathrm{tr}(\Sigma_2^{-1}\Sigma_1) - \log\frac{\det(\Sigma_1)}{\det(\Sigma_2)} - d\right)\,.$$

*Moreover, when $\Sigma_1 = \Sigma_2$,*

$$d(p, q) = \frac{1}{2}(\mu_2 - \mu_1)^\top\Sigma_2^{-1}(\mu_2 - \mu_1)\,.$$

*Proof.* The proof follows from the definitions of KL divergence and multivariate Gaussians. □

**Lemma 6.** *Fix $\mu_1 \in \mathbb{R}^d$, $\Sigma_1 \succeq 0$, $\mu_2 \in \mathbb{R}^d$, and $\Sigma_2 \succeq 0$. Then*

$$\mathcal{N}(x; \mu_1, \Sigma_1)\,\mathcal{N}(x; \mu_2, \Sigma_2) \propto \mathcal{N}(x; \mu, \Sigma)\,,$$

*where*

$$\mu = \Sigma(\Sigma_1^{-1}\mu_1 + \Sigma_2^{-1}\mu_2)\,, \quad \Sigma = (\Sigma_1^{-1} + \Sigma_2^{-1})^{-1}\,.$$

*Proof.* This is a classic result, which is proved as

$$\mathcal{N}(x;\mu_1,\Sigma_1)\mathcal{N}(x;\mu_2,\Sigma_2) \propto \exp\left[-\frac{1}{2}((x-\mu_1)^\top\Sigma_1^{-1}(x-\mu_1) + (x-\mu_2)^\top\Sigma_2^{-1}(x-\mu_2))\right]$$

$$\propto \exp\left[-\frac{1}{2}(x^\top\Sigma_1^{-1}x - 2x^\top\Sigma_1^{-1}\mu_1 + x^\top\Sigma_2^{-1}x - 2x^\top\Sigma_2^{-1}\mu_2)\right]$$

$$= \exp\left[-\frac{1}{2}(x^\top\Sigma^{-1}x - 2x^\top\Sigma^{-1}\Sigma(\Sigma_1^{-1}\mu_1 + \Sigma_2^{-1}\mu_2))\right]$$

$$\propto \exp\left[-\frac{1}{2}(x-\mu)^\top\Sigma^{-1}(x-\mu)\right] \propto \mathcal{N}(x;\mu,\Sigma)\,.$$

The neglected factors depend on constants $\mu_1$, $\mu_2$, $\Sigma_1$, and $\Sigma_2$. This completes the proof. $\qquad\square$

# B    Learning the Reverse Process

One property of our model is that $q(s_T) \approx \mathcal{N}(s_T;\mathbf{0}_d, I_d)$ when $T$ is sufficiently large [19]. Since $S_T$ has the same distribution in the reverse process $p$, $p$ can be learned from the forward process $q$ by simply reversing it. This is done as follows. Using the definition of the forward process in (4), Ho et al. [19] showed that

$$q(s_{t-1} \mid s_t, s_0) = \mathcal{N}(s_{t-1};\tilde{\mu}_t(s_t, s_0), \tilde{\beta}_t I_d) \tag{13}$$

holds for any $s_0$ and $s_t$, where

$$\tilde{\mu}_t(s_t, s_0) = \frac{\sqrt{\bar{\alpha}_{t-1}}\beta_t}{1-\bar{\alpha}_t}s_0 + \frac{\sqrt{\alpha_t}(1-\bar{\alpha}_{t-1})}{1-\bar{\alpha}_t}s_t\,, \quad \tilde{\beta}_t = \frac{1-\bar{\alpha}_{t-1}}{1-\bar{\alpha}_t}\beta_t\,, \quad \bar{\alpha}_t = \prod_{\ell=1}^{t}\alpha_\ell\,. \tag{14}$$

Therefore, the latent variable in stage $t-1$, $S_{t-1}$, is easy to sample when $s_t$ and $s_0$ are known. To estimate $s_0$, which is unknown when sampling from the reverse process, we use the forward process again. In particular, (4) implies that $s_t = \sqrt{\bar{\alpha}_t}s_0 + \sqrt{1-\bar{\alpha}_t}\varepsilon_t$, where $\varepsilon_t \sim \mathcal{N}(\mathbf{0}_d, I_d)$ is a standard Gaussian noise. This identity can be rearranged as

$$s_0 = \frac{1}{\sqrt{\bar{\alpha}_t}}(s_t - \sqrt{1-\bar{\alpha}_t}\varepsilon_t)\,.$$

To obtain $\varepsilon_t$, which is unknown when sampling from $p$, we learn to regress it from $s_t$ [19].

The regressor is learned as follows. Let $\varepsilon_t(\cdot;\psi)$ be a regressor of $\varepsilon_t$ parameterized by $\psi$ and $\mathcal{D} = \{s_0\}$ be a dataset of training examples. We sample $s_0$ uniformly at random from $\mathcal{D}$ and then solve

$$\psi_t = \arg\min_{\psi} \mathbb{E}_q\left[\|\varepsilon_t - \varepsilon_t(S_t;\psi)\|_2^2\right] \tag{15}$$

per stage. The expectation is approximated by sampled $s_0$. Note that we slightly depart from Ho et al. [19]. Since each regressor has its own parameters, the original optimization problem over $T$ stages decomposes into $T$ subproblems.

# C    Additional Experiments

This section contains four additional experiments.

## C.1    Additional Synthetic Problems

In Section 6.2, we show results for three hand-selected problems out of six. We report results on the other three problems in Figure 4. We observe the same trends as in Section 6.2.

## C.2    MNIST Experiment

The next experiment is on the MNIST dataset [31]. We start with learning an MLP-based multi-way classifier for digits and extract their $d = 8$ dimensional embeddings. These are used as features in

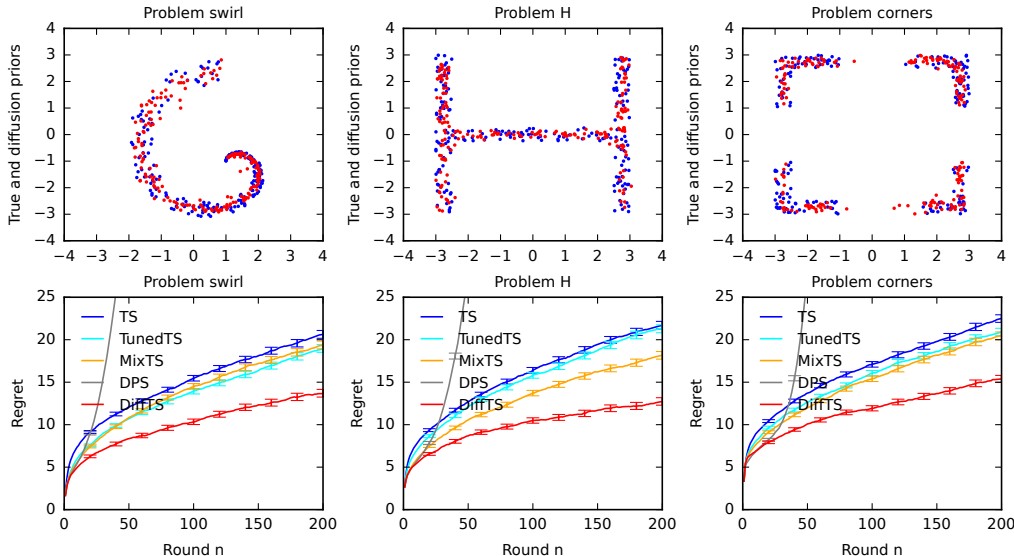

Figure 4: Evaluation of `DiffTS` on another three synthetic problems. The first row shows samples from the true (blue) and diffusion model (red) priors. The second row shows the regret of `DiffTS` and the baselines as a function of round $n$.

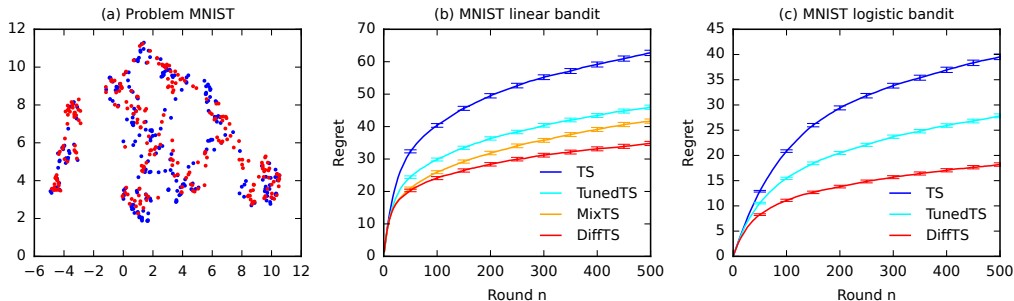

Figure 5: Evaluation of `DiffTS` on the MNIST dataset: (a) shows samples from the true (blue) and diffusion model (red) priors, (b) shows regret in the linear bandit, and (c) shows regret in the logistic bandit.

our experiment. We generate a distribution over model parameters $\theta_*$ as follows: (1) we choose a random positive label, assign it reward 1, and assign reward $-1$ to all other labels; (2) we subsample a random dataset of size 20, with $50\%$ positive and $50\%$ negative labels; (3) we train a linear model, which gives us a single $\theta_*$. We repeat this $10\,000$ times and get a distribution over $\theta_*$.

We consider both linear and logistic bandits. In both, the model parameter $\theta_*$ is initially sampled from the prior. In each round, $K = 10$ random actions are chosen randomly from all digits. In the linear bandit, the mean reward for a digit with embedding $x$ is $x^\top \theta_*$ and the reward noise is $\sigma = 1$. In the logistic bandit, the mean reward is $g(x^\top \theta_*)$, where $g$ is a sigmoid.

Our MNIST results are reported in Figure 5. We observe again that `DiffTS` has a lower regret than all baselines, because the learned prior captures the underlying distribution of $\theta_*$ well. We note that both the prior and diffusion prior distributions exhibit a strong cluster structure (Figure 5a), where each cluster represents one label.

## C.3 Ablation Studies

We conduct three ablation studies on the cross problem in Figure 2.

In all experiments, the number of samples for training diffusion priors was $10\,000$. In Figure 6a, we vary it from 100 to $10\,000$. We observe that the regret decreases as the number of samples increases,

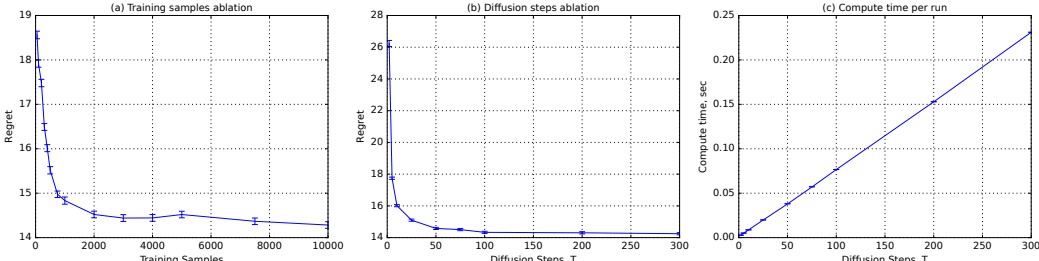

Figure 6: An ablation study of `DiffTS` on the cross problem: (a) we vary the number of samples for training the diffusion prior and report regret, (b) we vary the number of diffusion stages $T$ and report regret, and (c) we vary the number of diffusion stages $T$ and report computation time.

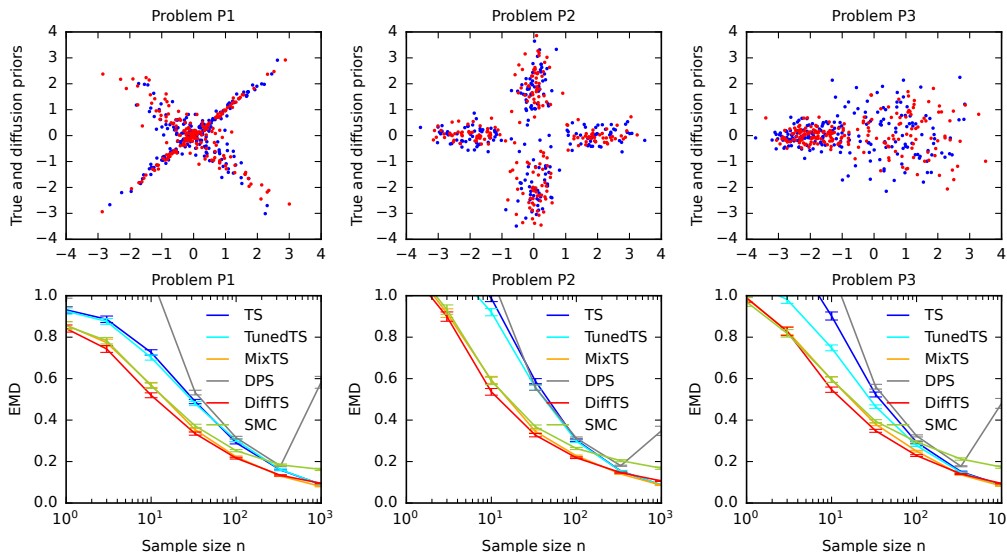

Figure 7: Evaluation on Gaussian mixture variants of the synthetic problems in Figure 2. The first row shows samples from the true (blue) and diffusion model (red) priors. The second row shows the earth mover's distance of `DiffTS` and baseline posteriors from the true posterior as a function of sample size $n$.

due to learning a better prior approximation. The trend stabilizes around $3\,000$ training samples. We conclude that the quality of the learned prior approximation has a major impact on regret.

In all experiments, the number of diffusion stages was $T = 100$. In Figure 6b, we vary it from $1$ to $300$ and observe its impact on regret. While the regret at $T = 1$ is high, it decreases quickly as $T$ increases. It stabilizes around $T = 100$, which we used in our experiments. In Figure 6c, we vary $T$ from $1$ to $300$ and observe its effect on the computation time of posterior sampling. The time is linear in $T$, as suggested in Section 4.2. The main contributor is the neural network regressor.

## C.4 Non-Bandit Evaluation

We use Gaussian mixture variants of the synthetic problems in Figure 2 for our non-bandit evaluation. The action in round $k$ is chosen uniformly at random (not adaptively). Since the priors are Gaussian mixtures, the true posterior distribution can be computed in a closed form using `MixTS` and we can measure the distance of posterior approximations from it. We use the *earth mover's distance (EMD)* between posterior samples from the true posterior and its approximation. We also considered the KL divergence. However, we could not apply it because the posteriors of `DiffTS` and `DPS` do not have analytical forms.

---
**Algorithm 4** DPS of Chung et al. [12].
---
1: **Input:** Model parameters $\tilde{\sigma}_t$ and $\zeta_t$

2: Initial sample $S_T \sim \mathcal{N}(\mathbf{0}_d, I_d)$
3: **for** stage $t = T, \dots, 1$ **do**
4:     $\hat{S} \leftarrow -\frac{\varepsilon_t(S_t; \psi_t)}{1 - \bar{\alpha}_t}$
5:     $\hat{S}_0 \leftarrow \frac{1}{\sqrt{\bar{\alpha}_t}}(S_t + (1 - \bar{\alpha}_t)\hat{S})$
6:     $Z \sim \mathcal{N}(\mathbf{0}_d, I_d)$
7:     $S_{t-1} \leftarrow \frac{\sqrt{\bar{\alpha}_{t-1}}\beta_t}{1 - \bar{\alpha}_t}\hat{S}_0 + \frac{\sqrt{\alpha_t}(1 - \bar{\alpha}_{t-1})}{1 - \bar{\alpha}_t}S_t + \tilde{\sigma}_t Z - \zeta_t \nabla \sum_{\ell=1}^{N}(y_\ell - \phi_\ell^\top \hat{S}_0)^2$

8: **Output:** Posterior sample $S_0$
---

We evaluate all methods from Figure 2. In addition, we implement a *sequential Monte Carlo (SMC)* sampler [16]. The initial particles are sampled uniformly at random from the prior. At each round, the particles are perturbed by a Gaussian noise. The standard deviation of the noise is initialized as a fraction of the observation noise and decays over time, as the posterior concentrates. The particles are weighted according to the likelihood of the observation in the round. Finally, we use normalized likelihood weights to resample the particles. The number of particles is 3 000 and we tune SMC to get good posterior approximations. The computational cost of SMC is comparable to DiffTS.

Our results are reported in Figure 7. We observe that DiffTS approximations are comparable to MixTS, which has an exact posterior in this setting. The second best performing method is SMC. Its approximations worsen as the sample size $n$ increases. DPS approximations also get worse as $n$ increases, which caused instability in Figure 2.

## D   Implementation of Chung et al. [12]

In our experiments, we compare to diffusion posterior sampling (DPS) with a Gaussian observation noise (Algorithm 1 in Chung et al. [12]). Our implementation is presented in Algorithm 4. The score is $\hat{S} = -\varepsilon_t(S_t; \psi_t)/(1 - \bar{\alpha}_t)$, where $\varepsilon_t(S_t; \psi_t)$ is a regression estimate of the forward process noise $\varepsilon_t$ in Appendix B. We set $\tilde{\sigma}_t = \sqrt{\tilde{\beta}_t}$, which is the same amount of noise as in our reverse process (Section 3). The term

$$\nabla \sum_{\ell=1}^{N}(y_\ell - \phi_\ell^\top \hat{S}_0)^2$$

is the gradient of the negative log-likelihood with respect to $S_t$.

As discussed in Appendices C.2 and D.1 of Chung et al. [12], $\zeta_t$ in DPS needs to be tuned for good performance. This is because $\nabla \sum_{\ell=1}^{N}(y_\ell - \phi_\ell^\top \hat{S}_0)^2$ grows with the number of observations, which causes instability. We also observed this in our experiments (Section 6.2). To make DPS work well, we follow Chung et al. [12] and set

$$\zeta_t = \frac{1}{\sqrt{\sum_{\ell=1}^{N}(y_\ell - \phi_\ell^\top \hat{S}_0)^2}}.$$

While this significantly improves the performance of DPS, it does not prevent failures. The fundamental problem is that gradient-based optimization is sensitive to the step size, especially when the optimized function is steep. Note that LaplaceDPS does not have any such hyper-parameter.

