# OpenReview forum: "Online Posterior Sampling with a Diffusion Prior"
_NeurIPS.cc/2024/Conference — NeurIPS 2024 poster_

### Official Review · Reviewer_AZGY · 2024-07-07

**Soundness:** 3
**Presentation:** 3
**Contribution:** 3
**Rating:** 7
**Confidence:** 3

**Summary:**

The paper studies online learning and proposes to approximate the updating prior with a diffusion model, rather than the more simple, less expressive Gaussian approximation. The main application is contextual bandits with a linear or GLM model. For the latter case, the authors derive a version of the IRLS algorithm (based on a Laplace approximation) that works with the diffusion prior. The authors also demonstrate the consistency of their approximating algorithm. The paper examines a variety of examples, and compares the proposed model to existing methods for contextual bandit.

**Strengths:**

The idea to extend the IRLS algorithm used for an updating Gaussian prior to a more sophisticated prior is well motivated and seems useful. The paper presents an actionable algorithm and the theoretical results are convincing. There are also some empirical results---both on toy examples and a benchmark data set---, which demonstrate the utility of the diffusion prior.

I also found the paper to be well written: it provides extensive context for existing approaches, notably the Laplace approximation and the IRLS algorithm. The proposed DiffTS then seems like a natural next step.

**Weaknesses:**

I picked up a few points, but I believe most of them can be addressed in the rebuttal.

I'm not entirely comfortable with the statements of Theorem 2 and 4, because they involve "$\approx$", which is not formally defined. For instance, a reader might wonder if the approximation in (6) is "as good" as the one on line 148... I understand what the authors mean, but formal statements need to be more precise. I recommend, as a fix, writing "Suppose that" and then equation (6) with an equality sign. The authors can then replace all the "$\approx$" with an "=", and state that the initial assumption does not hold in general; assuming it does is where the approximation comes into play.

In the experiments, the authors report the regret against round $n$. One thing that's not accounted for is that DiffTS requires more computation than its benchmarks. How much of a concern is this in practice? Does the added computation slow down model training or is the wait time dominated by generating a new round? Some discussion around this, potentially with an additional figure that reports regret vs time, could be helpful.

It was also not clear to me what where the tuning parameters of DiffTS. The authors claim it inherits the simplicity and efficiency of the Gaussian prior, however, determining T and picking a neural network architecture strike me as complications that limit the off-the-shelf use of the algorithms. I believe the authors could be more upfront about this; the paper would also be stronger with examination of those tuning choices across examples.

**Questions:**

If I understand correctly, computing the regret requires knowing the optimal $a_*$ and $\theta_*$. Are there evaluations which can be performed without oracle knowledge? What about diagnostics to assess the quality of the underlying Laplace approximation?

**Limitations:**

The authors address some of the limitations. See my comments above for additional limitations that should be addressed.

---

> ### Author Rebuttal · Authors · 2024-08-06
>
> We would like to thank the reviewer for positive feedback, which recognizes multiple contributions of our work. Our rebuttal is below. If you have any additional concerns, please reach out to us to discuss them.
>
> **Avoid $\approx$ in Theorems 2 and 4**
>
> A great comment. We agree that it is cleaner to state (6) as an assumption and then replace all $\approx$ with $=$.
>
> **Computation time and regret trade-offs of DiffTS**
>
> In our experiments, posterior sampling in DiffTS with $T$ stages is about $T$ times more computationally costly than posterior sampling with a Gaussian prior. We discuss this in lines 267-272 (Section 6.2). We plot the regret and sampling time as a function of $T$ in Figures 6b and 6c (Appendix C.3), respectively.
>
> **Diffusion model tuning**
>
> We have not done much tuning. In all experiments, the number of diffusion stages is $T = 100$ and the diffusion rate is set such that most of the signal diffuses. The regressor is a $2$-layer neural network and we learn it from $10000$ samples from the prior. These settings resulted in stable performance across all experiments. We plot the regret as a function of the number of training samples and $T$ in Figures 6a and 6b (Appendix C.3), respectively. When $T$ or the number of training samples is small, DiffTS performs similarly to posterior sampling with a Gaussian prior.
>
> **Non-bandit evaluation**
>
> We conduct an empirical evaluation in the non-bandit setting in the pdf attached to the **common rebuttal**. Please see it for more details.
>
> **Limitations of our method**
>
> Please see the **common rebuttal**.

---

> > ### Comment · Reviewer_AZGY · 2024-08-09
> > **Response to authors' rebuttals**
> >
> > I've read the authors' rebuttals.
> >
> > The authors have addressed my comments and I'm happy to maintain my good score.

---

### Official Review · Reviewer_1tRW · 2024-07-10

**Soundness:** 3
**Presentation:** 4
**Contribution:** 2
**Rating:** 5
**Confidence:** 3

**Summary:**

This paper introduces novel posterior sampling approximations tailored for diffusion model priors, specifically designed for use in contextual bandits and applicable to a broader range of online learning problems. The methods are developed for linear models and generalized linear models (GLMs), emphasizing enhanced stability and efficiency in environments where previous algorithms may exhibit instability and divergence. The paper provides the asymptotic consistency of these approximations. Empirically, the performance of these approximations is evaluated on contextual bandit problems, showcasing their capability to manage uncertainties inherent in these settings effectively.

**Strengths:**

This paper proposes approximate posterior sampling algorithms for contextual bandits with a diffusion model prior. Unlike previous research using a Gaussian prior, it addresses the instability and divergence issues. This paper has a well-structured presentation and elucidates background knowledge such as linear models, GLM, and diffusion models. Claims are supported by either theoretical proofs and numerical studies. The proofs are technically sound based on the reviewer's examination.

**Weaknesses:**

It seems to the reviewer the key idea in this paper is to use the diffusion model instead of a Gaussian distribution as a prior. The proofs seem to be standard and similar to those based on a Gaussian prior. The main weaknesses of the paper are its lack of novelty and the absence of significant technical challenges.

Some format issues: p2 l49, p4 Fig. 1, l121, and l118.

**Questions:**

Can the author explain why the diffusion model prior solves the instability and divergence issues?

What other benefits could the diffusion model prior bring?

What other approaches could be employed beyond Laplace approximation?

What are the technical difficulties in the paper, especially the analysis?

**Limitations:**

Discussion on limitations seems to be missing or need to be more explicit.

---

> ### Author Rebuttal · Authors · 2024-08-06
>
> We would like to thank the reviewer for valuable feedback and praising our execution. Our rebuttal is below. If you have any additional concerns, please reach out to us to discuss them.
>
> **Q1: Score issue in posterior sampling**
>
> There may be a misunderstanding. The diffusion prior does not solve any instability. We address previous issues with posterior sampling and a diffusion model prior.
>
> Prior works on posterior sampling in diffusion models sampled using the score of the posterior probability (Section 7). The score depends on the gradient of the log-likelihood $\nabla \log p(h | \theta)$, which grows linearly with the number of observations $N$ and causes instability. We combine $p(h | \theta)$ with the conditional prior, in each stage of the diffusion model, using the Laplace approximation. The resulting conditional posterior concentrates at a single point as $N \to \infty$ and is easy to sample from, although its gradient goes to infinity. Our main technical contribution is an efficient and asymptotically consistent implementation of this solution, using a stage-wise Laplace approximation with diffused evidence.
>
> **Q2: Benefit of diffusion model priors**
>
> The main benefit of diffusion models is that they can learn complex prior distributions from data. These distributions then serve as representations of prior knowledge.
>
> **Q3: Other approaches to posterior sampling**
>
> The most popular approach is DPS of Chung et al. (2023). The key idea in DPS is to sample from a diffusion model posterior by adding the score of the likelihood of observations to the diffusion model prior. We describe this approach in Appendix D and compare to it empirically in Section 6. We also mention several other approaches in Section 7. All of them rely on the score of the likelihood $\nabla \log p(h \mid \theta)$ and thus become unstable as the number of observations $N$ increases.
>
> **Q4: Technical challenges**
>
> We do not just replace a Gaussian prior with a diffusion model prior in classic posterior sampling. We propose posterior sampling with a diffusion model prior that can be implemented efficiently and asymptotically consistently using a stage-wise Laplace approximation with diffused evidence. Please see the **common rebuttal** for more details on technical challenges.
>
> **Limitations of our method**
>
> Please see the **common rebuttal**.

---

> > ### Comment · Reviewer_1tRW · 2024-08-13
> > **Reply to authors**
> >
> > Thanks for the authors' responses. I'll keep my score.

---

### Official Review · Reviewer_YMJY · 2024-07-14

**Soundness:** 3
**Presentation:** 3
**Contribution:** 3
**Rating:** 6
**Confidence:** 3

**Summary:**

The paper presents approximate posterior sampling methods for contextual bandits with a diffusion prior. A key weakness of existing methods designed to work on noisy data is that rely on using the score function is that it becomes unstable as the number of observations grows. Instead, the authors propose to use closed-form analytic updates to merge prior and evidence distributions in the diffusion chain. This works for linear models and can be extended to generalized linear models with the Laplace approximation. This framework is applied in the context of contextual bandits for synthetic and real datasets and performs better than other diffusion model alternatives.

**Strengths:**

The problem of Thompson sampling with complex multi-modal distribution is a significant one with a lot of impact. The use of identifying (or approximating with) Gaussians to enable closed-form updating in a diffusion setting is very compelling and results in a simple solution that seems to work well in contextual bandits. The Laplace approximation is a natural choice to move beyond simple linear model settings. The writing and development of the method and theorems was mostly well executed.

**Weaknesses:**

A few notes about correctness/presentation:
(1) it is not true that the posterior of \theta_* is a product of two multivariate Gaussians (as stated below Eq 3) it is instead proportional to that product.
(2) there seems to be a non-sequitur on L82-83 when Laplace approximation is introduced: Laplace does not require the prior to Gaussian, it approximates the posterior as a Gaussian.

The introduction of IRLS with corresponding algorithm in Sec 2.2. seemed a bit out of place and I wonder if it could not be delayed until a little bit later in the method section (in part because it is not explained why one couldn't just use autodiff to find the MAP solution as is now standard).

It was not clear which of the proofs are standard results and which are novel (e.g. Lemma 1 looks like Bayes theorem).

**Questions:**

In Algorithm 2, the mean and covariance functions for all S_{0:T} are functions of the history which can be of variable size, how does the network take this aspect into account?

**Limitations:**

The idea seems quite general but was only considered in a contextual bandits setting, is there a reason why you couldn't compare the samples in a non-bandit setting to other methods e.g. VI, SMC, MCMC?

---

> ### Author Rebuttal · Authors · 2024-08-06
>
> We would like to thank the reviewer for detailed feedback, and praising our contributions and execution. Our rebuttal is below. If you have any additional concerns, please reach out to us to discuss them.
>
> **Some claims are imprecise**
>
> The reviewer is right that the posterior in (3) is only proportional to the product of the prior and likelihood. They are also right that the Laplace approximation does not require a Gaussian prior, although this is a very common setting.
>
> **Novelty in proofs**
>
> Please see the **common rebuttal**.
>
> **How do the mean and covariance in Algorithm 2 change with history?**
>
> They are weighted sums of the prior quantities, which are represented by a neural network and independent of history $h$, and empirical quantities. As an example, $\hat{\Sigma}_t(h)$ in line 3 is computed in (8) in Theorem 2. It is the inverse of a weighted sum of the prior $\Sigma_t^{-1}$ and empirical $\bar{\Sigma}^{-1}$ precisions. The latter is defined right after (3) in Section 2.1.
>
> **Non-bandit evaluation**
>
> We conduct an empirical evaluation in the non-bandit setting in the pdf attached to the **common rebuttal**. Please see it for more details.

---

> > ### Comment · Reviewer_YMJY · 2024-08-12
> > **thank you**
> >
> > Thank you for the rebuttal including the common responses. I maintain my current score of weak accept.

---

### Official Review · Reviewer_r4r4 · 2024-07-15

**Soundness:** 3
**Presentation:** 2
**Contribution:** 2
**Rating:** 6
**Confidence:** 2

**Summary:**

The authors propose an algorithm for sampling from a generalised linear model posterior where the prior is defined through a diffusion model. This is achieved by utilizing the Laplace approximation, and is shown to be asymptotically consistent. This model and inference scheme is then applied to contextual bandits, where their performance is demonstrated on a variety of synthetic and real world applications.

**Strengths:**

I am not familiar with Bandits or Diffusion models but the paper is interesting and the experiments in the context of contextual bandits seem convincing.

**Weaknesses:**

1)	This paper reads like a paper of two halfs. The first half is proposing a new sampling algorithm, and the second is applying this to contextual bandits. I would have like to have seen more evidence / discussion on how well this algorithm works in general for diffusion models.  For example, do you know how well your method works on the experimental setup of Chung et al [12] ?

2)	The math at times can be a bit difficult to parse (however this could be due to not being familiar with this field). Minor things like using bold symbols for vectors/matrices may help here.

**Questions:**

1)	In Thm 2 equation 6 reads $p(h | s_t) = E_{p(s_0 | s_t}[p(h|s_0)] \approx p(h | s_t / sqrt(\alpha_t))$. Which seems like the assumption is $s_t \approx  s_t / sqrt(\alpha_t)$, why do you want to make this assumption and why is justified?

2)	Why does $\nabla log p(h|\theta)$ grow linearly in N?

3)	Does relying on the score of likelihood only impact the application for contextual bandits or does this also impact these methods generally?

4)	Is there any error caused/limitation in using the Laplace approximation?

**Limitations:**

They discuss that they have performed regret analysis. However they do discuss limitations of the proposed sampling algorithm.

---

> ### Author Rebuttal · Authors · 2024-08-06
>
> We would like to thank the reviewer for valuable feedback. Our rebuttal is below. If you have any additional concerns, please reach out to us to discuss them.
>
> **Experimental setup of Chung et al. (2023)**
>
> Chung et al. (2023) experiment with computer vision problems. Their algorithm is unstable in these problems without tuning, and they discuss it in their Appendices C.2 and D.1. We focused on online learning as the first step because a suboptimal model of uncertainty, even after tuning, is unlikely to perform well. See the performance of DPS in Figures 2 and 4, and our discussion in Appendix D. We plan to experiment with vision and video models in our future work.
>
> **Q1: Approximation (6) in Theorem 2**
>
> We assume that $s_0 = s_t / \sqrt{\bar{\alpha}_t}$, where $s_0$ is a clean sample and $s_t$ is the corresponding diffused sample in stage $t$. This approximation is motivated by the observation that under the forward process, $s_t = \sqrt{\bar{\alpha}_t} s_0 + \sqrt{1 - \bar{\alpha}_t} \tilde{\varepsilon}_t$ for any $s_0$, where $\tilde{\varepsilon}_t \sim \mathcal{N}(\mathbf{0}_d, I_d)$ is a standard Gaussian noise. See lines 166-170 in Section 4.3. The result of our approximation is that the likelihood becomes a function of scaled $s_t$, and can be easily combined with the conditional prior distribution, which is also a function of $s_t$.
>
> **Q2: Why does $\nabla \log p(h | \theta)$ grow linearly in $N$?**
>
> Because the history $h$ involves $N$ observations. As an example, in Assumption 1, the likelihood is $p(h \mid \theta) \propto \exp[- \sigma^{-2} \sum_{\ell = 1}^N (y_\ell - \phi_\ell^T \theta)^2]$.
>
> **Q3: Is the score issue general?**
>
> Yes. This is a general issue of relying on $\nabla \log p(h | \theta)$ when $N$ is large. Chung et al. (2023) tune their gradient step to mitigate this. See our Appendix D for more details.
>
> **Q4: Does the Laplace approximation have an error?**
>
> Yes. We discuss this in lines 166-175 (Section 4.3). The good news is that the error vanishes as the number of observations increases. This is what we prove in Theorem 3.
>
> **Limitations of our method**
>
> Please see the **common rebuttal**.

---

> > ### Comment · Reviewer_r4r4 · 2024-08-12
> >
> > Thank you for your response. I will maintain my already positive score.

---

### Official Review · Reviewer_Vag9 · 2024-07-16

**Soundness:** 4
**Presentation:** 3
**Contribution:** 3
**Rating:** 6
**Confidence:** 2

**Summary:**

The paper introduces new posterior sampling approximations for contextual bandits with diffusion model priors, allowing the capability to handle complex distributions beyond traditional Gaussian priors. It contributes by developing efficient sampling algorithms, proving their asymptotic consistency, and validating their effectiveness on synthetic and empirical contextual bandit problems.

**Strengths:**

The paper is overall well-written, providing a clear background and enhancing accessibility for a broad audience. It advances prior work by extending Thompson sampling with a diffusion model prior from K-armed bandits to a more general setting of contextual bandits, which broadens the application scope. The theoretical claims, such as asymptotic consistency, are supported by sound arguments. The experiment are also comprehensive. Furthermore, the authors also mentioned the potential in extensions beyond GLM, suggesting broader applications.

**Weaknesses:**

The paper briefly mentions extending the work of Hsieh et al. [22], who proposed Thompson sampling with a diffusion model prior for K-armed bandits, to contextual bandits. However, the discussion lacks depth and specificity. A more detailed comparison is needed to highlight the unique contributions of this paper. For example, the difference between the posterior sampling approximations, and the difference between the theoretical analysis.

The experiment sections (Figures 4 and 5) lack explanations on the metrics, which are crucial for understanding the variability and reliability of the results. It should be clear whether the error bar is the standard deviation or the standard error of the mean.

**Questions:**

As mentioned in the "Weakness" section, a somewhat detailed summary that clarifies the differences between this work and related works, particularly the work of Hsieh et al. [22], would help understand the unique contributions of this paper.

Clarity would be enhanced by more explanations on the roles of the reverse process and forward process in posterior sampling.

**Limitations:**

I recommend that the authors include a more detailed discussion of the limitations of this work.

---

> ### Author Rebuttal · Authors · 2024-08-06
>
> We would like to thank the reviewer for detailed and positive feedback. Our rebuttal is below. If you have any additional concerns, please reach out to us to discuss them.
>
> **Differences from Hsieh et al. (2023)**
>
> There are multiple differences:
>
> * The posterior approximation in Hsieh et al. (2023) is for scalars (individual arms). Our approximation is for vectors (model parameters).
> * The approximations are different. In stage $t$, Hsieh et al. (2023) sample from the conditional prior and the diffused empirical mean distribution in stage $t$. Then they take a weighted average of the samples. We sample only once, from the posterior distribution obtained by combining the conditional prior in stage $t$ and likelihood. Based on this, Hsieh et al. (2023) can be viewed as a non-contextual variant of our method, where posterior sampling is done by weighting samples from the prior and empirical distributions.
> * Hsieh et al. (2023) do not analyze their approximation.
>
> **Metrics in Figures 4 and 5**
>
> The regret is defined as in Figures 2 and 3 (Section 6). The mathematical definition is given in line 213 (Section 5). All error bars are standard errors.
>
> **Limitations of our method**
>
> Please see the **common rebuttal**.

---

> > ### Comment · Reviewer_Vag9 · 2024-08-12
> >
> > Thanks for the authors' responses. The discussions on the limitations and the prior work now look more comprehensive to me. I'm happy to keep my positive score.

---

### Author Rebuttal · Authors · 2024-08-06

We wanted to thank all reviewers for positive reviews and recognizing our contributions. There were three common comments that we want to address jointly: limitations of the work, technical challenges, and a non-bandit evaluation.

**Limitations of our method**

We point out limitations of our work throughout the paper. However, the reviewers could not always find them. To address this issue, we plan to have a dedicated paragraph for limitations in Conclusions. The main limitations of our approach are:

* **Computational cost:** Posterior sampling in DiffTS with $T$ stages is about $T$ times more computationally costly than posterior sampling with a Gaussian prior. We say this in lines 162-164 (Section 4.2) and validate it experimentally in lines 267-272 (Section 6.2). We plot the sampling time as a function of $T$ in Figure 6c (Appendix C.3).
* **Diffusion model prior tuning**: In all experiments, the number of diffusion stages is $T = 100$ and the diffusion rate is set such that most of the signal diffuses. The regressor is a $2$-layer neural network and we learn it from $10000$ samples from the prior. These settings resulted in stable performance across all experiments without tuning. We plot the regret as a function of the number of training samples and $T$ in Figures 6a and 6b (Appendix C.3), respectively. When $T$ or the number of training samples is small, DiffTS performs similarly to posterior sampling with a Gaussian prior.

**Technical challenges**

Prior works on posterior sampling in diffusion models sampled using the score of the posterior probability (Section 7). The score depends on the gradient of the log-likelihood $\nabla \log p(h | \theta)$, which grows linearly with the number of observations $N$ and causes instability. We combine $p(h | \theta)$ with the conditional prior, in each stage of the diffusion model, using the Laplace approximation. The resulting conditional posterior concentrates at a single point as $N \to \infty$ and is easy to sample from, although its gradient goes to infinity. Our main technical contribution is an efficient and asymptotically consistent implementation of this solution, using a stage-wise Laplace approximation with diffused evidence.

Lemma 1 can be derived using basic rules of probability and we state it in Appendix A.1. All other claims, including efficient posterior derivations (Theorems 2 and 4) and asymptotic consistency (Theorem 3), rely on our proposed approximation of clean samples by scaled diffused samples. The most challening part of the analysis is Theorem 3, which analyzes an asymptotic behavior of a chain of $T$ random vectors, which depend on each other.

**Non-bandit evaluation**

We use Gaussian mixture variants of the synthetic problems in Figure 2 for our non-bandit evaluation. The action in round $k$ is chosen uniformly at random (not adaptively). Since the priors are Gaussian mixtures, the true posterior distribution can be computed in a closed form using MixTS and we can measure the distance of posterior approximations from it. We use the *earth mover's distance (EMD)* between posterior samples from the true posterior and its approximation. We also considered KL divergence, but this one required analytical forms of posterior approximations, which are not available in DiffTS and DPS.

We compare all methods from Figure 2. In addition, we implemented a sequential Monte Carlo (SMC) sampler. The initial particles are chosen uniformly at random from the prior. At each next round, the particles are perturbed by a Gaussian noise. The standard deviation of the noise is initialized as a fraction of the observation noise and decays over time, as the posterior concentrates. The particles are weighted according to the likelihood of the observation in the round. Finally, normalized likelihood weights are used to resample the particles. We tune SMC to get the best approximations. We use $3000$ particles. For this setting, the posterior sampling times of SMC and DiffTS are comparable.

Results of our experiments are reported in Figure 7 in the attached pdf. We observe that the quality of DiffTS approximations is similar to MixTS, which has an exact posterior in this setting. The second best performing method is SMC. The quality of its approximations worsens as the sample size $n$ increases. The quality of DPS approximations also worsens as $n$ increases, which caused instability in Figure 2.

---

### Author Response · Authors · 2024-08-12

Dear reviewers,

Thank you for the reviews and taking our rebuttal into account when evaluating the paper. The reviewer-author discussion period will end soon (August 13 EoD AoE). If you have any additional concerns about the paper, we would be happy to address them in the next 2 days.

Sincerely,

Authors

---

### Decision · Program_Chairs · 2024-09-25

**Decision:**

Accept (poster)

**Comment:**

This paper has borderline scores. The reviewers all emphasize good-quality writing. The main weakness noted by the lowest-scoring review is that the techniques used are relatively standard, especially for a paper which is more computational / empirical / similar in nature, compared to other papers in this area that are doing pure theorem-proof-based regret analysis. Still, even that reviewer writes that the work is technically sound in terms of the numerics, which are arguably one of the more important aspects in a paper of this kind. On basis of this thinking, and discussion, I am inclined to accept.